# NMDA-receptor-dependent plasticity in the bed nucleus of the stria terminalis triggers long-term anxiolysis

Christelle Glangetas[1,2,*], Léma Massi[3,*], Giulia R. Fois[4,*], Marion Jalabert[5,6], Delphine Girard[1,2,4], Marco Diana[7], Keisuke Yonehara[3], Botond Roska[3], Chun Xu[3], Andreas Lüthi[3], Stéphanie Caille[8,9] & François Georges[1,2,4]

Anxiety is controlled by multiple neuronal circuits that share robust and reciprocal connections with the bed nucleus of the stria terminalis (BNST), a key structure controlling negative emotional states. However, it remains unknown how the BNST integrates diverse inputs to modulate anxiety. In this study, we evaluated the contribution of infralimbic cortex (ILCx) and ventral subiculum/CA1 (vSUB/CA1) inputs in regulating BNST activity at the single-cell level. Using trans-synaptic tracing from single-electroporated neurons and in vivo recordings, we show that vSUB/CA1 stimulation promotes opposite forms of in vivo plasticity at the single-cell level in the anteromedial part of the BNST (amBNST). We find that an NMDA-receptor-dependent homosynaptic long-term potentiation is instrumental for anxiolysis. These findings suggest that the vSUB/CA1-driven LTP in the amBNST is involved in eliciting an appropriate response to anxiogenic context and dysfunction of this compensatory mechanism may underlie pathologic anxiety states.

[1] Université de Bordeaux, Interdisciplinary Institute for Neuroscience, UMR 5297, F-33076 Bordeaux, France. [2] Centre National de la Recherche Scientifique, Interdisciplinary Institute for Neuroscience, UMR 5297, F-33076 Bordeaux, France. [3] Friedrich Miescher Institute for Biomedical Research, Maulbeerstrasse 66, 4058 Basel, Switzerland. [4] Centre National de la Recherche Scientifique, Neurodegeneratives Diseases Institute, UMR 5293, F-33076 Bordeaux, France. [5] Université de la Méditerranée UMR S901, F-13009 Aix-Marseille 2, France. [6] INMED, F-13009 Marseille, France. [7] 'G. Minardi' Cognitive Neuroscience Laboratory, Department of Chemistry and Pharmacy, University of Sassari, 07100 Sassari, Italy. [8] Université de Bordeaux, Institut de Neurosciences Cognitives et Intégrative d'Aquitaine, BP31, F-33076 Bordeaux, France. [9] Centre National de la Recherche Scientifique, UMR 5287-Institut de Neurosciences Cognitives et Intégrative d'Aquitaine, F-33076 Bordeaux, France. * These authors contributed equally to this work. Correspondence and requests for materials should be addressed to F.G. (email: francois.georges@u-bordeaux.fr).

Anxiety is a physiological negative emotion that triggers a state of alert to possible threat and promotes survival. The bed nucleus of the stria terminalis (BNST) belongs to a neuronal network of interconnected limbic regions and exerts a pivotal role in the expression of anxiety both in humans[1,2] and in rodent models[3,4]. Previous studies using electrical[5], lesioning[6,7], pharmacological[8] or optogenetic[3,4] manipulations targeting directly specific BNST nuclei or cell types have been informative but have not been able to determine precisely how different BNST inputs influence anxiety. It has been shown that the ventral subiculum/CA1 (vSUB/CA1) and infralimbic cortex (ILCx) massively projects to the BNST[9]. However, the integrative properties of BNST neurons at the single-cell level have never been explored. One hypothesis is that cortical and hippocampal information is processed at the single-cell level in BNST to trigger anxiolysis.

One of the most likely neural mechanisms underlying persistent anxiety is long-lasting plasticity in the neuronal network[10]. Probably owing to their anatomically convergent and segregated excitatory inputs from the infralimbic Cortex (ILCx) and the vSUB/CA1 and their high content in stress-related neuromodulators, BNST circuits display morphological or synaptic plastic adaptations in response to stress and anxiety[11–13]. *Ex vivo* studies in slice have correlated plasticity at glutamate synapses within the BNST with alterations in anxiety levels[11], but it remains unclear how the BNST integrates diverse inputs to modulate anxiety-related behaviors. Synaptic homeostasis seems to be a crucial process to compensate a long-lasting enhancement in signal transmission and maintain the stability of neuronal activity[14,15]. Recent computational modeling of synaptic plasticity have shown that the homeostatic processes that control the network stability are supported by the interaction of homosynaptic plasticity with heterosynaptic plasticity[16]. To unravel the integrative properties of BNST neurons at the single-cell level and the mechanism of their plastic changes in response to specific input stimulation, we combined conventional tracing, trans-synaptic tracing from single-electroporated neurons, *in vivo* single-cell recordings, pharmacological and behavioral techniques. Here, we demonstrate: (1) that both vSUB/CA1 and ILCx converge on the same amBNST neurons; (2) that *in vivo* stimulation of the vSUB/CA1 promotes an N-methyl-D-aspartic acid or N-methyl-D-aspartate (NMDA)-dependent long term potentiation (LTP) at the vSUB/CA1-amBNST synapses (LTP$_{vSUB/CA1}$), associated with an NMDA-independent long-term depression (LTD) at the ILCx-amBNST synapses (LTD$_{ILCx}$); and (3) that the induction of *in vivo* NMDA-R-dependent plasticity in the amBNST triggers long-term changes of anxiety state.

## Results

**BNST neurons are connected to both vSUB/CA1 and ILCx inputs.** It has been shown that the BNST could integrate information from the vSUB/CA1 or the ILCx (refs 9,12,17–19). However, it was unknown whether these inputs were integrated separately by different BNST neurons or if the same BNST neuron integrates both inputs. To address this question, we first injected a retrograde tracer, the cholera-toxin-B subunit (CTb), into the BNST, and confirmed that the BNST receives strong innervations from the vSUB/CA1 and the ILCx (Supplementary Fig. 1a–c). We then injected two different anterograde tracers, Phaseolus-vulgaris-leucoagglutinin (PHAL) and biotinylated-dextran-amine (BDA) in the vSUB/CA1 and in the ILCx, respectively (Fig. 1a,b). We used the Fox3 protein as a neuronal marker in the BNST. Using confocal imaging, we showed strong convergences of ILCx and vSUB/CA1 terminal fibers on the same Fox3-positive amBNST neuron (Fig. 1c). By using *in vivo* electrophysiology in anesthetized rats, we investigated whether amBNST neurons were controlled by both vSUB/CA1 and ILCx projections at the single-

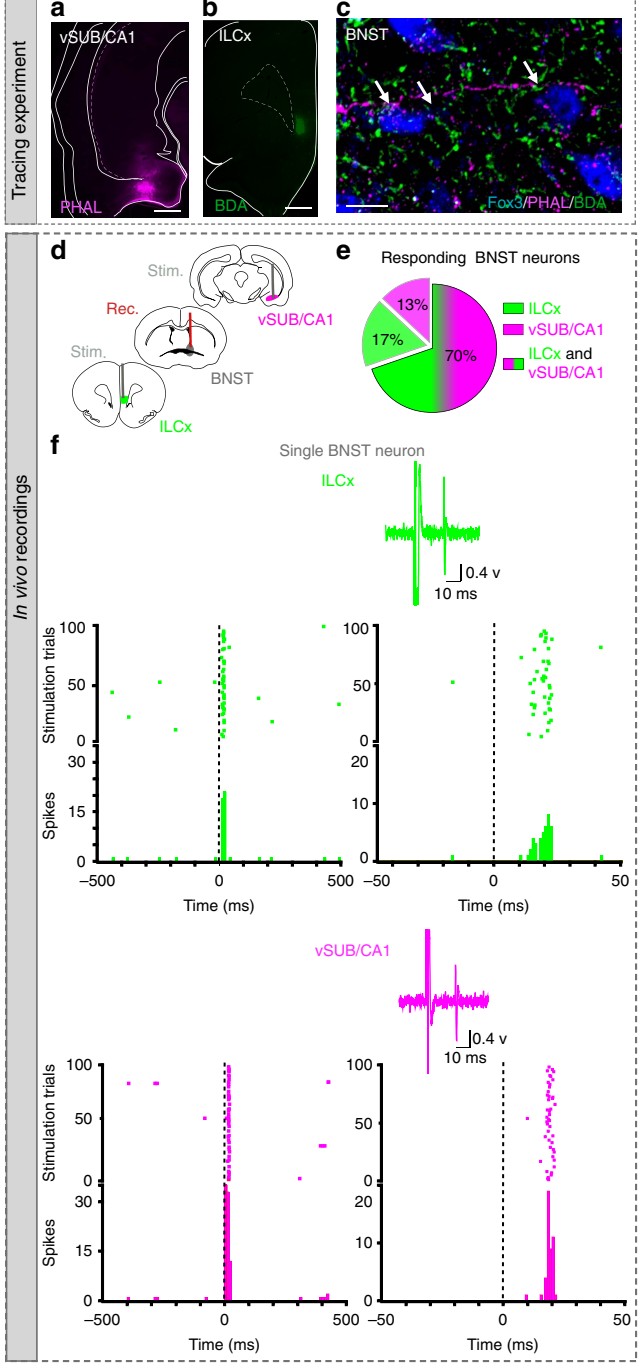

**Figure 1 | Characterization of vSUB/CA1-ILCx-amBNST pathways.** (**a,b**) Injection sites: anterograde tracers (PHAL, BDA) were respectively injected in the vSUB/CA1 (**a**, magenta) and the ILCx (**b**, green). Scale, 0.5 mm. (**c**) Immunofluorescence confocal images showing strong convergences (white arrows) of vSUB/CA1 (magenta) and ILCx (green) terminals, on amBNST neuron (Fox3 labelling, blue). Scale bar, 10 μm. (**d**) Experimental protocol. (**e**) Percentage of amBNST neurons responding to the stimulation of the ILCx, vSUB/CA1 or both. (**f**) Typical PSTHs ( + raster) showing respective excitatory responses of a same amBNST neuron to ILCx (top) and vSUB/CA1 (bottom) stimulation. Stimulus: t0 (gray lines). PSTH with 500 ms scale and Bin width, 5 ms (left) and PSTH of the same neuron with 50 ms scale and bin width, 1 ms on the right. Insets: representative traces.

cell level. When we tested the evoked responses of all recorded amBNST neurons to the stimulation of the ILCx and the vSUB/CA1, we observed that 85.2% of the recorded neurons responded

to the stimulation of one of these inputs (Fig. 1d). We quantified that 70% of these responding neurons responded to both ILCx and vSUB/CA1 stimulation, (Fig. 1e,f), whereas only 17% and 13% responded to either ILCx or vSUB/CA1 stimulation, respectively (Fig. 1e). The basal firing rate of the responding amBNST neurons was $0.7 \pm 0.2$ Hz. The onset of the evoked responses for the ILCx-amBNST pathway was $6.94 \pm 0.59$ ms and $9.75 \pm 0.77$ ms for the vSUB/CA1-amBNST pathway ($P < 0.01$). Durations of evoked excitatory responses did not differ between these two pathways (ILCx-amBNST: $12.06 \pm 1.06$ ms; vSUB/CA1-amBNST: $12.78 \pm 1.16$ ms). The intensity of stimulation used to trigger evoked responses in the amBNST was in the same range between the two inputs ($0.2–1$ mA). To further dissect connectivity of amBNST inputs onto single-amBNST neurons, we performed *in vivo* single-cell electroporation of amBNST neuron followed by retrograde monosynaptic tracing with pseudotyped rabies virus strategy (Fig. 2a,b)[20]. Five days after electroporation, retrogradely infected neurons were detected in the ILCx and in the vSUB/CA1 (Fig. 2b). Unfortunately, starter cells were not detected in the amBNST, most probably because rabies virus infection will eventually induce cytotoxicity in infected neurons[21]. Together, our findings confirm that a large population of

amBNST neurons was synaptically controlled, at the single-cell level, by both vSUB/CA1 and ILCx inputs.

**HFS_vSUB/CA1 promotes *in vivo* input-specific LTD and LTP.** To unravel the integrative properties of BNST neurons at the single-cell level, we combined *in vivo* single-cell recordings and pharmacological approaches. Given that vSUB/CA1 neurons fire at hundreds of Hertz in basal condition (Supplementary Fig. 2a–c), we first assessed the impact of a high-frequency stimulation (HFS; Supplementary Fig. 2d,e) of the vSUB/CA1 (HFS_vSUB/CA1), on the ability of the vSUB/CA1-amBNST and ILCx-amBNST synapses to undergo plasticity (Fig. 3a–d). This HFS_vSUB/CA1, originally described by Abraham et al.[22], induced an extremely long-lasting ($>8$ days), robust and stable LTP in the projection from vSUB/CA1 to medial prefrontal cortex (mPFC)[23]. Here, we established that HFS_vSUB/CA1 triggers input-specific neuroplastic changes in the BNST (Fig. 3). After 30–40 min, HFS_vSUB/CA1, amBNST neurons exhibited a 42.1% increase in the vSUB/CA1-evoked spike probability (LTP_vSUB/CA1) together with a 47.4% decrease in the ILCx-evoked spike probability (LTD_ILCx; $F_{(1.15)} = 35.97$, $P < 0.0001$ for the HFS_vSUB/CA1 protocol × input-stimulated interaction; $F_{(1.15)} = 80.60$, $P < 0.0001$ for the effect of the input stimulated, Fig. 3c–e). No change of the basal firing rate of amBNST neurons was observed before and after the HFS_vSUB/CA1 ($2.74 \pm 1.03$ Hz versus $1.91 \pm 0.74$ Hz; $P = 0.401$). We next delivered HFS_vSUB/CA1 in the presence of intra-BNST infusion of the NMDA-R antagonist AP5 ($100 \mu M$). Under these conditions, HFS_vSUB/CA1 failed to elicit LTP_vSUB/CA1 ($41.79 \pm 13.16\%$ of baseline, $n = 10$, $P < 0.001$; $F_{(1.15)} = 33.89$, $P < 0.001$ for the effect of AP5 treatment × effect of HFS_vSUB/CA1 protocol interaction; $F_{(1.15)} = 28.60$, $P < 0.0001$ for the effect of the AP5 treatment; Fig. 3f). Under the same conditions, HFS_vSUB/CA1 still induced LTD_ILCx ($2.76 \pm 2.76\%$ of baseline, $n = 5$, $P < 0.001$; $F_{(1.13)} = 9.68$, $P < 0.01$ for the effect of AP5 treatment × effect of HFS_vSUB/CA1 protocol interaction; $F_{(1.13)} = 97.77$, $P < 0.0001$ for the effect of HFS_vSUB/CA1 protocol; $F_{(1.13)} = 6.90$, $P < 0.05$ for the effect of AP5 treatment, Fig. 3g). In addition, HFS protocol applied in the ILCx (HSF_ILCx) was ineffective to trigger plasticity in the ILCx nor in the vSUB/CA1 (Supplementary Fig. 3).

***In vivo* LTP in the amBNST triggers anxiolysis.** Finally, we conducted behavioral experiments to test our hypothesis that vSUB/CA1-driven NMDA-receptor-dependent LTP in the amBNST triggers anxiolysis (timeline Fig. 4a), as both regions are implicated in anxiety-related behaviors[24,25]. To test whether HFS_vSUB/CA1 promotes anxiolytic effect in basal but also in anxiogenic situation, we used two complementary anxiety-tests based on the innate aversion of rodents to brightly illuminated areas, with the light–dark test (Fig. 4b–f) or to open spaces with the elevated plus maze (EPM; Fig. 4e,g,h)[26]. In the light–dark test, rats exposed to HFS_vSUB/CA1 spent more time in the light compartment in basal situation compared with SHAM group (percentage of time in light; HFS_vSUB/CA1 group, $36.8 \pm 3.8\%$; SHAM group, $27.5 \pm 2.7\%$; $P < 0.05$, Fig. 4c). Intra-BNST AP5 infusion prevented the anxiolytic-like effect induced by HFS_vSUB/CA1. AP5 injection into the BNST decreased the % of time spent in light only in the HFS_vSUB/CA1 group ($P < 0.05$ for the effect of AP5 injection, Fig. 4c,d). Neither novelty-induced locomotor activity (Supplementary Fig. 4a) nor circadian rhythms of general activity (Supplementary Fig. 4b) were altered after HFS_vSUB/CA1 or after manipulating the activity on the NMDA receptors in the BNST, thereby reinforcing the specific role of homeostatic plasticity on behavioral anxiety. There was no significant difference in the number of transitions between groups (light–dark test: SHAM/aCSF: $15.20 \pm 0.60$ transitions, SHAM/AP5: $16.50 \pm 1.50$ transitions,

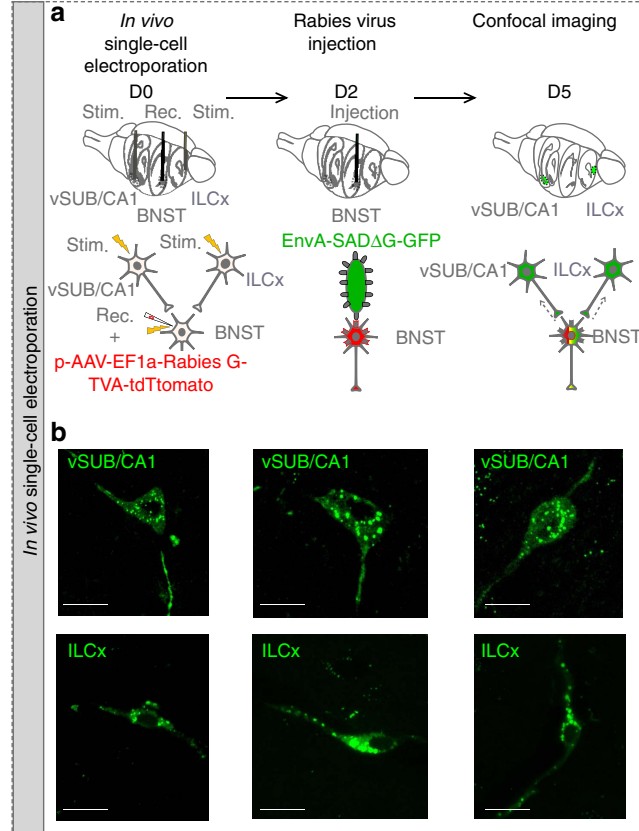

**Figure 2 | Trans-synaptic tracing from single-electroporated amBNST neurons.** (**a**) Experimental protocol of *in vivo* single-cell electroporation of amBNST neuron coupled with a recombinant monosynaptic retrograde-pseudotyped rabies virus strategy. A single-amBNST neuron, responding to both ILCx and vSUB/CA1 stimulations, was electroporated *in vivo* with a plasmid coding for the rabies glycoprotein, the TVA receptor and a fluorescent marker (pAAV-EF1a-RabiesG-TVA-tdTomato). Two days later, a pseudotyped rabies (EnvA-SAD○G-GFP) was injected into the amBNST and will infect only the electroporated amBNST neuron. (**b**) Confocal images of vSUB/CA1 (top) and ILCx (bottom) GFP-positive neurons (scale bar, 10 μm).

$HFS_{vSUB/CA1}$/aCSF: $17.20 \pm 2.20$ transitions, $HFS_{vSUB/CA1}$/AP5: $16.30 \pm 1.50$ transitions; $P > 0.05$) (Supplementary Fig. 4c). In the light–dark test or in the EPM, the light serves as an anxiogenic stimulus[27]. To create an anxiogenic situation, the lighting was increased from 560 Lux to 1,230 Lux, and rats were restrained for 5 min before the light–dark test and the lighting was set-up at 260 Lux in open arms for the EPM (Fig. 4e)[28]. In the light–dark test, as expected, when rats where submitted to anxiogenic situation, they spent less time in the light compared with the SHAM group in basal situation (SHAM group in basal situation, $27.5 \pm 2.7\%$; SHAM group in anxiogenic situation, $10.16 \pm 3.1\%$,

$P < 0.005$, Fig 4c,f). In addition, when rats are exposed to anxiogenic situation in the light–dark test or EPM (Fig. 4e–h), $HFS_{vSUB/CA1}$ has still an anxiolytic effect (light–dark test: SHAM group in anxiogenic situation, $10.16 \pm 3.1\%$; $HFS_{vSUB/CA1}$ group in anxiogenic situation, $19.76 \pm 4.2\%$; $P < 0.05$, Fig. 4f; EPM: percentage of time in the open arms, SHAM group: $14.95 \pm 2.51\%$; $HFS_{vSUB/CA1}$ group: $23.38 \pm 3.65\%$; $P < 0.05$, Fig. 4h). No differences were observed between groups for the number of transitions in the light–dark test or EPM (light–dark test: SHAM group: $9.90 \pm 0.89$ transitions; $HFS_{vSUB/CA1}$ group: $11.29 \pm 1.60$ transitions; $P > 0.05$, Supplementary Fig. 4d; EPM: SHAM group: $16.14 \pm 1.47$ transitions; $HFS_{vSUB/CA1}$ group: $19.29 \pm 1.96$ transitions; $P > 0.05$, Supplementary Fig. 4e). Taken together, these data indicate that (1) $HFS_{vSUB/CA1}$ decreases the steady-state anxiety level through a NMDA-receptor-dependent mechanism in the BNST (2) acute $HFS_{vSUB/CA1}$ diminishes the anxiety induced by an anxiogenic situation.

## Discussion
Using tract-tracing, trans-synaptic tracing from single-electroporated neurons and *in vivo* recordings, we report that the majority of neurons of the anteromedial BNST integrates information from the vSUB/CA1 and the ILCx at the single-cell level. Considering the differences in the straight-line distances between the ILCx-BNST ($\approx 4$ mm) and vSUB/CA1-BNST ($\approx 6$ mm), the 3 ms difference in the latency to the onset of the stimulation response between the two inputs is probably not supported by a difference in conduction velocity, but by the length of the axonal projections. We next found that $HFS_{vSUB/CA1}$ triggers *in vivo* an evoked spike potentiation ($LTP_{vSUB/CA1}$), which requires the activation of NMDA-Rs in the BNST. This HFS protocol was efficient to trigger plasticity in the amBNST when applied in the vSUB/CA1, but not in the ILCx (Supplementary Fig. 3). This is probably due to the fact that vSUB/CA1 is one of the few major output structures of the hippocampal formation and transmits learning and memory-related signals in a high-frequency bursting mode (Supplementary Fig. 2) repeated at a low frequency ($0.5$–$2$ Hz)[29]. A pioneering study in the hippocampus demonstrated that changes in synaptic strengths affect network activity and shape neuronal integration in an input-specific manner[30]. Here, we report that in response to an excessive activity of the vSUB/CA1-amBNST inputs ($LTP_{vSUB/CA1}$), amBNST neurons at the single-cell level down-regulate the efficacy of their ILCx-amBNST inputs ($LTD_{ILCx}$) and maintain their basal activity stable. This interaction of homosynaptic plasticity ($LTP_{vSUB/CA1}$) with heterosynaptic plasticity ($LTD_{ILCx}$) occurring at the single-cell level at two amBNST excitatory synapses could be considered as homeostatic plasticity[16]. In fact, these neuroplastic changes correspond to a form of homeostasis necessary to maintain

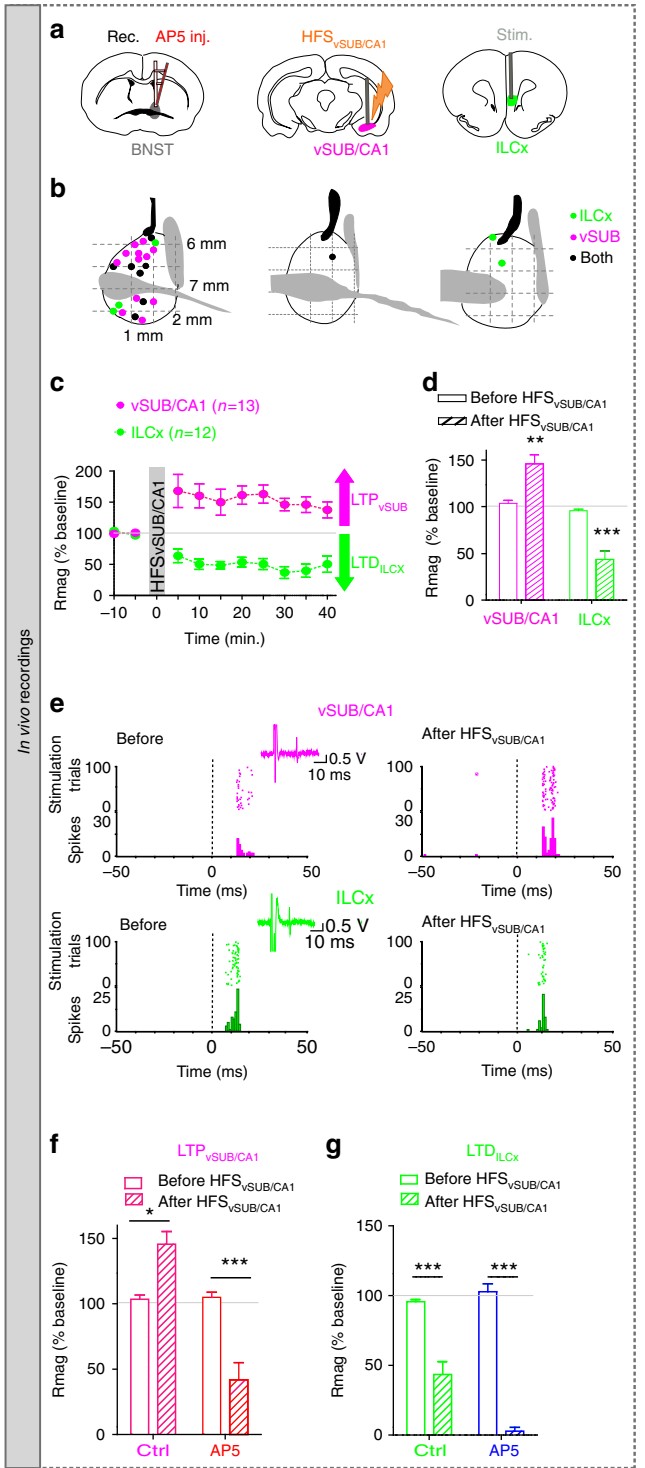

**Figure 3. | *In vivo* input-specific opposite plasticity in the amBNST.**
(**a**) Experimental protocol. (**b**) Cartography of recording sites in the BNST. Neurons which have been tested for the ILCX are represented in green, those tested for the vSUB are in magenta and those tested for both inputs are in black. (**c,d**) Kinetic (**c**) and quantification (**d**) of the mean percentage change ($\pm$ s.e.m.) in vSUB/CA1 (magenta) and ILCx (green) evoked spike probability, normalized to the baseline, after $HFS_{vSUB/CA1}$. Rmag, excitatory response magnitude. (**e**) Typical PSTHs ($+$ rasters) illustrating responses of a same single-BNST neuron before (left) and after (right) $HFS_{vSUB/CA1}$ on the vSUB/CA1-amBNST pathway (top) and on the ILCx-amBNST pathway (bottom). Stimulus: t0 (gray lines). Bin width: 1 ms. Traces in insets. (**f,g**) Quantification of the mean percentage change ($\pm$ s.e.m.) in the vSUB/CA1 spike probability (**f**) or in ILCx spike probability (**g**) in control condition or after AP5 infusion in the amBNST.

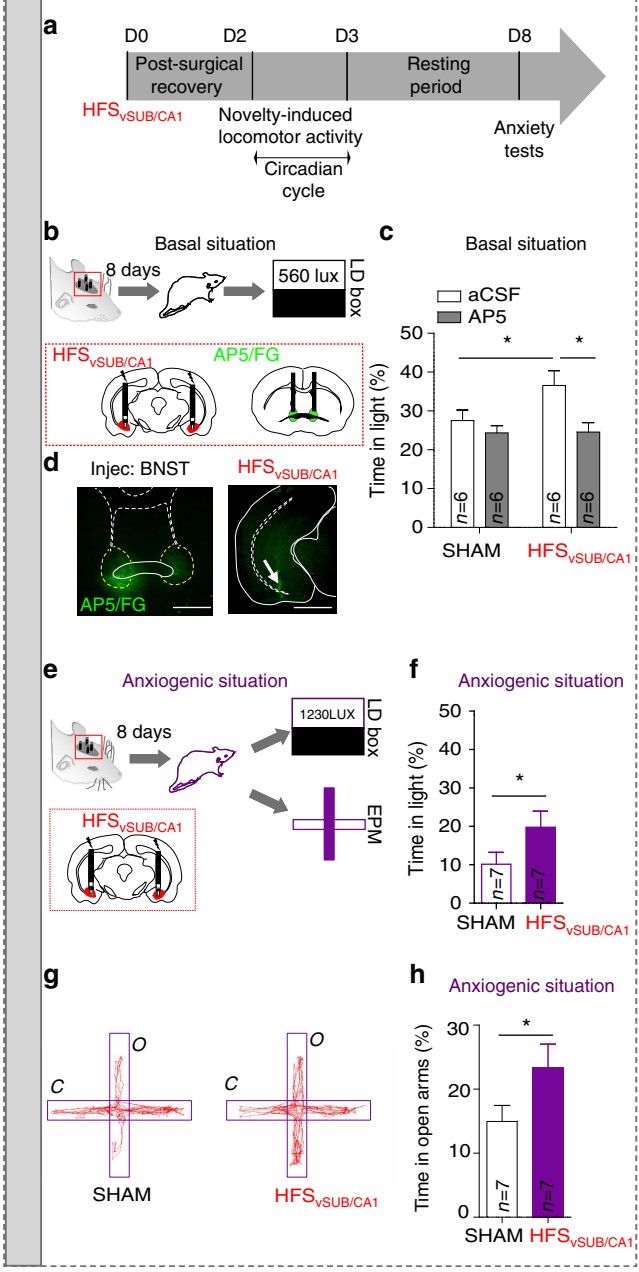

**Figure 4 | *In vivo* NMDA-R antagonist infusion in BNST neurons blocks anxiolytic effect induced by HFS$_{vSUB/CA1}$.** (**a**) Timeline of the behavioral experiments and experimental protocol. (**b**) Experimental protocol in basal situation. (**c**) Histograms showing the performances at the light–dark box test after AP5 (grey) or vehicle in the BNST (aCSF, white) followed by HFS$_{vSUB/CA1}$ or SHAM manipulation in basal situation. (**d**) Histological control. Scale bar, 1.0 mm. (**e**) Experimental protocol in anxiogenic situation. (**f**) Histograms showing the performances at the light–dark box test followed by HFS$_{vSUB/CA1}$ (purple) or SHAM manipulation (white) in anxiogenic situation. (**g**) Example trace of a SHAM rat (left) and a HFS$_{vSUB/CA1}$ treated rat (right) performances at the EPM (**h**) Histograms showing the percentage of time spent in the open arms in the EPM in the SHAM group and in HFS vSUB/CA1 group in anxiogenic situation.

stability in the amBNST, in response to the strengthening of the vSUB/CA1-amBNST synapses (LTP$_{vSUB/CA1}$) associated with a negative feedback at the ILCx-amBNST synapses (LTD$_{ILCx}$). Regardless of the cellular explanation, the functional effect of LTD$_{ILCx}$ is complementary to that of LTP$_{vSUB/CA1}$, that is it

optimizes the signal-to-noise ratio by reinforcing the functional weight of the recently potentiated synapses. Interestingly, in presence of AP5 in the amBNST, the direction of plasticity at the vSUB/CA1-amBNST inputs elicited by HFS$_{vSUB/CA1}$, was switch from an LTP$_{vSUB/CA1}$ to an LTD$_{vSUB/CA1}$ (Fig. 3f). One possibility is that AP5, locally infused in the amBNST, only partially blocks the NMDA receptors, and this partial blockade reverses the direction of plasticity elicited HFS$_{vSUB/CA1}$ (refs 31,32). Another unexpected result was that the intra-amBNST blockade of NMDA-Rs potentiates LTD$_{ILCx}$ elicited by HFS$_{vSUB/CA1}$ (Fig. 3g). One possibility is that is that HFS$_{vSUB/CA1}$ triggers concomitant activation of NMDA and metabotropic glutamate receptors in amBNST neurons, leading to a more profound LTD in the presence of AP5 (ref. 33). Further experiments are necessary to determine the molecular mechanisms by which in the absence of NMDA-Rs stimulation, HFS$_{vSUB/CA1}$ triggers LTD$_{vSUB/CA1}$ and potentiates LTD$_{ILCx...}$. Finally, we can not exclude that HFS$_{vSUB/CA1}$ also triggers plasticity in the mPFC[23] or the basolateral amygdala[34], but we provide behavioral evidence that vSUB/CA1-driven NMDA-R-dependent LTP in the amBNST triggers anxiolytic-like effects. This is in line with pioneer studies showing that changing the activity in the amBNST has a direct impact on the perception of aversive contextual stimuli[35] or production of stress hormones[5]. In fact, here we have demonstrated, using two different anxiety assays that HFS$_{vSUB/CA1}$ induced an anxiolytic effect in basal situation but also in anxiogenic situation (Fig. 4). Together, these data support the conclusion that the amBNST plays a crucial role in integrating and sending information related to anxiety[9,36]. Previous studies have shown that anxiety is controlled by multiple circuits in the brain, many of which share robust and reciprocal connections with the BNST[4,37]. These circuits include projections from the basolateral nucleus of the amygdala (BLA) to the ventral hippocampus[38], from BLA to the central nucleus of the amygdala (CeA)[39], from the ventral hippocampus to the medial prefrontal cortex (mPFC)[40], from mPFC to BLA[41] and from BLA to BNST[4]. Our anatomical and functional chara-cterization of the vSUB/CA1-amBNST projection on a circuit and synaptic level furthers the understanding of the role played by amBNST in the modulation of anxiety[4,37]. In conclusion, we show that in response to HFS$_{vSUB/CA1}$, homeostasis in amBNST neurons is guaranteed at the single-cell level by an NMDA-R-dependent up-scaling of the vSUB/CA1-amBNST synapses associated with an NMDA-R independent down-regulation of the efficacy of its ILCx-amBNST inputs (LTD$_{ILCx}$; Supplementary Fig. 5). Together these findings elucidate the molecular targets that contribute to changes in synaptic functions in the amBNST, and highlight important future directions where manipulation of inputs to the amBNST using opto- or chemogenetic tools may be critical for changing network output, physiological manifestations of anxiety and anxiety-associated disorders[42].

## Methods

**Animals.** Male sprague Dawley rats (275–300 g; 10 weeks old; Elevage Janvier, France) were used. Rats were housed three or four per cage under controlled conditions (22–23 °C, 40% relative humidity, 12 h light/dark illumination cycle; lights on from 07:00 hours to 19:00 hours), were acclimatized to laboratory conditions 1 week before the experiment, with food and water *ad libidum*. All procedures were conducted in accordance with the European directive 2010-63-EU and with approval from Bordeaux University Animal Care and Use Committee (N° 50120205-A).

**Surgery.** *In vivo* electrophysiology. Stereotaxic surgeries for electrophysiology, tract-tracing and for light–dark test experiments were performed under 1.0–1.2% isoflurane (in 50% air/50% O$_2$; 1 l min$^{-1}$) anesthesia[43]. Stimulation electrodes, recording pipettes or injection pipettes were, respectively, inserted into the ILCx (+3.0 mm/bregma, 0.5 mm/midline, 4.5 mm/brain surface), the vSUB/CA1 (−6.0 mm/bregma, 5.1 mm/midline, 7.1 mm/brain surface), the amBNST (0.0 mm/bregma, 1.3 mm/midline, 6–7.5 mm/brain surface).

**Electrical stimulation of the ILCx and the vSUB/CA1.** Bipolar electrical stimulation of the vSUB/CA1 and ILCx was conducted with a concentric electrode (Phymep, France) and a stimulus isolator (500 μs duration, 0.2–2 mA; Digitimer, UK). Baseline was recorded for 10 min ($2 \times 100$ pulses; 0.5 Hz). To induce LTP, high-frequency stimulation protocol was performed in the vSUB/CA1 (HFS$_{vSUB/CA1}$) at the same intensity used for baseline (0.2–1 mA). To avoid the confounding effect of epilepsy driven behavioral changes occurring in awake freely moving animals[23], HFS was delivered in anesthetized animals. HFS$_{vSUB/CA1}$ consisted in 50 trains (500 pulses at 400 Hz, 250 μs duration pulse) presented as bursts of five trains. The frequency of the five trains was 1 Hz. Each burst of five trains was presented five times at 1 min interval (Supplementary Fig. 2).

**amBNST recordings.** A glass micropipette (tip diameter, 1–2 μm; 10–15 MΩ) filled with a 2% pontamine sky blue solution in 0.5 M sodium acetate was lowered into the amBNST. The extracellular potential was recorded with an Axoclamp-2B amplifier and filter (300 Hz/0.5 Hz). Spikes were collected online (CED 1401, SPIKE 2; Cambridge Electronic Design; UK). During electrical stimulation of the ILCx or vSUB/CA1, cumulative peristimulus time histograms (PSTH, 5 ms bin width) of amBNST activity were generated for each neuron recorded.

**Pharmacological treatment.** For local delivery of 100 μM AP5, double barrel pipettes were used[43]. For behaviour, a mixture of 180 nl of AP5 (100 μM) and 0.2% Fluorogold (to mark the injection site) was injected bilaterally in the BNST.

**Histology.** At the end of each recording experiment, the recording pipette placement was marked with an iontophoretic deposit of pontamine sky blue dye ($-20$ μA, 30 min). To mark electrical stimulation sites, $+50$ μA was passed through the stimulation electrode for 90 s. After, brains were removed and snap-frozen in a solution of isopentane stored at $-80$ °C.

**Plasmid solution.** A plasmid encoding for the rabies glycoprotein, the avian virus receptor TVA and a fluorescent marker (tdTomato) was used in this study. For recording followed by electroporation experiments, the electrode was filled with the plasmid (pAAV-EF1a-G-TVA-tdTomato, 17.5 ng μl$^{-1}$) diluted in standard intracellular solution.

***In vivo* single-cell electroporation.** Single-cell electroporation was performed as described previously[44]. After recording amBNST neurons responding to both ILCx and vSUB/CA1 stimulations, they were electroporated with a solution containing a plasmid DNA (pAAV-EF1a-G-TVA-tdTomato). We applied $-10$ V square-pulses delivered at 50 Hz for 1 s. Only one cell per rat was electroporated. After 2 days, an EnvA pseudotyped G-deleted rabies virus (EnvA-SAD○G-GFP) was injected into the amBNST. After 5 days, the electroporation protocol, rats were killed for immunohistological experiments.

**Tract-tracing method.** Tracer injections were performed as described previously[43] with the following modifications. For retrograde tracing, 30 nl of 0.5% CTb (Sigma Aldrich, France) were infused by pressure into the amBNST ($n = 5$). Animals received a single iontophoretic injection of a 2.5% solution of an anterograde tracer PHAL (Vector Laboratories; UK) in the vSUB/CA1 ($n = 5$). Animals received a single iontophoretic injection of a 2.5% solution of an anterograde tracer BDA in the ILCx ($n = 3$). Animals were allowed to survive 7–14 days.

**Immunohistochemical methods.** Immunocytochemical detection was performed by standard light, confocal microscopy as previously described[43]. Rats were perfused transcardially (4% paraformaldehyde solution). Sections were incubated (overnight per 4 °C) with rabbit or goat anti-PHAL primary antibody (1/2,000, Cliniciences, France), rabbit anti-Fox3 primary antibody (1/1,000; Abcam, France), goat anti-CTb primary antibody (1/10,000, List biological laboratories, USA). For fluorescence microscopy, after washing sections were incubated overnight at 4 °C with a donkey anti-rabbit secondary antibody (labelling of PHAL, 1/1,000, Invitrogen, France, Alexa 568), donkey anti-goat secondary antibody (labelling of CTb,1/1,000, Jacskon Immunoresearch, UK, Alexa 647), biotinylated antibody coupled with an Alexa 488 (labelling of BDA, Vector laboratories) or donkey anti-goat secondary antibody (labelling of PHAL,1/1,000, Invitrogen, France, alexa 568), donkey anti-rabbit secondary antibody (labelling of Fox3, 1/1,000, Jacskon immunoresearch, UK, Alexa 647). Sections were washed and then mounted in Vectashield medium (Vector Laboratories), coverslipped and imaged on a laser scanning confocal microscope. The confocal microscope was a Leica TCS-SP5-STED on an inverted stand DMI6000 (Leica Microsystems, Germany), using objectives HCX Plan APO CS $\times 10$ dry NA 0.40, HCX Plan Apo CS $\times 20$ multi-immersion NA 0.70, HCX Plan Apo CS $\times 63$ oil NA 1.40. For conventional confocal microscopy the lasers used were Argon (458 nm, 476 nm, 488 nm, 496 nm, 514 nm) Green Helium-Neon 561 nm, Orange Helium-Neon 594 nm and Red Helium-Neon 633 nm. The scanning was done with a resonant scanner (8,000 Hz–16,000 Hz). The microscope was equipped with 5 internal photomultiplier tube (PMT), two of them are hybrid detectors, 2 NDD PMT,

2 APD and 1 PMT Trans. Photomicrographs were taken and displayed using Image J to perform z stack images.

**Elevated plus maze.** One week later SHAM and HFS stimulations, it has been performed an EPM test. The apparatus is in plexiglass, with four elevated arms arranged in a cross-like disposition, it consisted in two opposite enclosed arms and two open arms, having at their intersection a central square platform which gave access to any of the four arms. Each arm is 50 cm; the wall for the closed arm is 40 cm height. All floor surfaces were grey and the open arms were under an anxiogenic illumination of 260 Lux. In brief, the rats were placed individually in the central square of the EPM facing an open arm and then allowed to start exploring the maze freely during 5 min test. Video recordings were analysed offline using video tracking software (Videotrack from Viewpoint, Lyon, France). The following behaviours were measured: time spent in each compartments and number of transitions between compartment. The percentage of time spent in open arms was evaluated to assess anxiety.

**Light–dark test.** Eight days after the application of HFS in the vSUB/CA1 in a new group of anesthetized rats, we performed the light–dark test. The test lasted 5 min and was performed in a two-compartment box ($40 \times 40 \times 35$ cm) with two equal compartments that limit exploratory behavior. An aperture enabled the rats to pass from one compartment to the other. One was completely enclosed by black opaque plastic sides, with a lid of the same material, while the other was white, had no lid, and was brightly illuminated (560 lux). At the start of the experiment a rat was placed in the center of the lighted box with its head facing the wall opposite to the door. The latency for the first emergence from the dark to the light compartment, time spent in each compartment and frequency of explorations of the light compartment were recorded. Time in a zone was considered when the animal placed its four paws in that zone. Rats performed only one time the light–dark test.

A separate group of rats was exposed to anxiogenic situation in order to increase the anxious state of the rats. The intensity of the lighting was increases from 560 Lux to 1,230 lux, and animals were restrained for 5 min, and after 2 min of recovery were exposed to the light–dark test[45]. The same behavioral parameters were measured in anxiogenic situation as in normal situation.

**Data analysis.** For *in vivo* electrophysiological experiments, cumulative PSTHs of aBNST activity were generated during stimulation of ILCx or vSUB/CA1. Excitatory magnitudes ($R_{mag}$ values) were normalized for different levels of baseline impulse activity. Baseline activity was calculated on each PSTH, during the 500 ms preceding the stimulation. For each PSTH, $R_{mag}$ values for excitation were calculated according to: excitation $R_{mag} = $ (number of spikes in excitatory epoch)$-$ (mean number of spikes per baseline bin $x$ number of bins in excitatory epoch). The cortical or hippocampal excitation strength onto amBNST neurons was determined as the amount of current needed to obtain a 50% spike probability for ILCx-evoked responses or vSUB/CA1-evoked responses ($R_{mag}$ ranging from 30 to 60). Results are expressed as mean ± s.e.m. For statistic, two-group comparisons were achieved using Student's *t*-tests or Mann–Whitney when necessary. For multiple comparisons, values were subjected to a two-way Anova followed if significant by Bonferroni post hoc tests or to Kruskall–Wallis Anova for the behavioral part.

**Data availability.** The data that support the findings of this study are available from the corresponding author upon reasonable request.

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

## Acknowledgements

The microscopy was done in the Bordeaux Imaging Center a service unit of the CNRS-INSERM and Bordeaux University, member of the national infrastructure France BioImaging. We thank Philippe Legros for technical support and precious advices for confocal images acquisition. We thank Francis Chaouloff and Mario Carta for scientific discussion. We also thank for Ed Callaway for generously sharing materials. This work was supported by grants from Centre National de la Recherche Scientifique (CNRS), University of Bordeaux, Agence Nationale de la Recherche (ANR-12-BSV4-0022 to F.G.), by LABEX BRAIN ANR-10-LABX-43 and Region Aquitaine and by a NARSAD Young Investigator Grant from the Brain & Behavior Research Foundation to Léma Massi.

## Author contributions

C.G., S.C. and F.G. designed the experiments. L.M., C.X. and A.L. designed and performed the *in vivo* single-cell electroporation experiments. K.Y. and B.R. made the plasmid used for electroporation. C.G., M.J., G.R.F., S.C., L.M. and D.G. collected and analyzed the data. C.G., M.D., S.C., L.M., A.L., G.R.F. and F.G. wrote the manuscript. All authors discussed the results and commented the manuscript.

## Additional information

**Competing financial interests:** The authors declare no competing financial interests.

**How to cite this article**: Glangetas, C. *et al.* NMDA-receptor-dependent plasticity in the bed nucleus of the stria terminalis triggers long-term anxiolysis. *Nat. Commun.* **8,** 14456 doi: 10.1038/ncomms14456 (2017).

**Publisher's note**: 

