## [Peer Review File · Nature Communications]

Reviewers' comments:

Reviewer #1 (Remarks to the Author):

Glangetas et al set out to answer a question that is very important, but often ignored in the field's pursuit of linking synapses, circuits and behavior. Specifically, how does one reconcile changes in synaptic strength at one set of synapses with stability of firing in target neurons? Does this require some type of adaptive response at other synapses? Here they address this question beautifully using a series of elegant and challenging approaches. First they show that single neurons in BNST receive inputs from both ventral subiculum/CA1 neurons and neurons in the infralimbic cortex. Then, they show that if one set of synapses (vsub/CA1-BNST) is potentiated in vivo, then the other synapses (iLCtx-BNST) are depressed. This, they suggest indicates a type of homeostatic plasticity. This part of the manuscript is well done, but I do have some questions that, if addressed appropriately, would help strengthen this part of the paper. The authors then ask whether this plasticity has a behavioral consequence. In this part of the manuscript they fall short. In fact, there is little support for the authors assertions that, "Interplay between an NMDA-receptor-dependent homosynaptic long-term potentiation and an NMDA-receptor-independent heterosynaptic long-term depression is instrumental for anxiolysis."

Major Concerns

1) Although the authors show that potentiation at the vsub/CA1-BNST connection does promote anxiolysis (albeit it is quite modest) they fail to show any link between homeostatic changes in BNST and this behavior. Can the authors provide any evidence to support the idea that both potentiation at the vsub/CA1 pathway AND depression in the iLCx pathway are necessary for anxiolysis?

2) If my understanding is correct, the HFS and subsequent assessment of plasticity were done in anaesthetized animals. The behavior, clearly required awake behaving animals.

a. Can the authors be certain that plasticity observed in anaesthetized animals can be faithfully reproduced in awake, behaving animals?

b. Why was light-dark test conducted 8 days after HFS in vSUB/CA1? When examining plasticity in this pathway, the authors test for changes in spike probability, thirty to forty minutes after HFS. Do the authors have evidence that LTP lasts for 8 days after HFS?

3) The rationale for the HFS protocol is unclear. The reference provided (Abraham et al, 1993) does not specifically provide evidence that this is physiologically relevant. I understand that protocols to induce plasticity vary widely across preparations and brain regions, so it is important the authors provide stronger support for their contention that vsub/CA1 neurons fire at hundreds of Hertz.

4) It appears that the 'LTP' reported in the vsub/CA1 pathway is NMDAR-dependent. Interestingly, it appears that HFS in the presence of AP5 unmask an LTD. What is the explanation for this? Also, why is the LTD in the iLCx projection even more profound in the presence of AP5?

5) The increase in time spent in the light following HFS is quite modest - from 28% to 32%. If the authors are to make a strong case for 'LTP' in this pathway being anxiolytic, additional behavioral tests are warranted.

6) I'm curious about why there is not a reciprocal form of plasticity if HFS is delivered to the iLCx, rather than vsub/CA1. The authors mention this briefly, but do not discuss it further.

Reviewer #2 (Remarks to the Author):

This is an interesting report demonstrating the anatomical convergence and physiological interaction of ventral hippocampal and prefrontal cortical inputs on single BNST neurons. The authors find that high-frequency stimulation of ventral hippocampal afferents in the BNST causes an NMDA-receptor dependent long-term potentiation, as well as an NMDA-independent homosynaptic depression at IL afferents. NMDA receptor antagonists in the BNST dampened the anxiolytic effects of ventral hippocampal stimulation. These are intriguing findings, but there are some important questions concerning the results as described:

1) Which region of the BNST exhibited converging hippocampal and prefrontal afferents? FigS1 suggests convergence in areas both dorsal and ventral to the anterior commissure--where were the recordings obtained? The BNST is quite diverse--did the physiological properties described differ according to the region from which the recordings were made?

2) Ventral hippocampal stimulation might have induced plasticity in other efferents including the PFC and BLA. In fact, individual neurons in the VH project to both regions and have been implicated in regulating fear. Although the effects of stimulation were dampened by BNST APV, is it possible that these other circuits play a role?

3) Tetanic stimulation of the ventral hippocampus was found to have an anxiolytic effect in the light:dark box, which was attenuated by NMDA antagonists in the BNST. Much of the lesion literature suggests that VH activation is anxiogenic. How do the authors account for the anxiolytic effects of tetanic stimulation of VH relative to the larger literature that would suggest the opposite would be true?

Reviewer #3 (Remarks to the Author):

This manuscript provides evidence for a direct excitatory projection from the infralimbic cortex (ILCx) and ventral subiculum / CA1 (vSUB/CA1) to individual neurons in the BNST. Data is provided showing that high frequency stimulation (HFS) of the vSUB/CA1 results in an NMDA-dependent homosynaptic long term potentiation (LTP) and an NMDA-independent heterosynaptic long term depression (LTD) of the ILCx input. Interestingly, HFS of the vSUB/CA1 causes a reduction in anxiety-like behavior that is blocked by AP5 in the BNST, suggesting the LTP of the vSUB/CA1 pathway is necessary for anxiolysis. This study provides data that significantly furthers our knowledge of the circuitry of the BNST, however, the representation and interpretation of the data is significantly lacking in both clarity and depth.

Major Concerns:

1. A major concern with this manuscript is the lack of a direct cause-effect relationship between the LTP and LTD due to the HFS and the behavioral changes observed. The authors should be cautious not to overstate their conclusions. The data shown suggests that HFS of the vSUB/CA1 pathway caused a decrease in anxiety-like behavior more than a week later. However, the authors directly relate the LTP and LTD that occurs immediately after the HFS with the change in behavior eight days later with no data to support this conclusion. Additionally, a second behavioral anxiety test would strengthen the evidence that HFS of the vSUB/CA1 can have long-term effects on anxiety-like behavior in rats.

2. The location of the recordings in the BNST needs to be made clearer. There is significant evidence that the different regions of the anterior BNST, including the oval, anterolateral, anteromedial, and

ventral BNST, have different effects on anxiety-like behavior in rodents (Dunn, 1987; Haugler et al, 2013; Kim et al, 2013). If recordings were from throughout the anterior BNST, it is unclear how LTP of the vSUB/CA1 synapses could cause a reduction in anxiety-like behavior when optogenetic and chemogenetic inhibition of some of these neurons is known to decrease anxiety-like behavior (Kim et al, 2013; Pleil et al, 2015). The proportion of neurons that respond to each stimulation should be described in terms of their location in the BNST. For example, if the majority of recordings, but not all, were performed in the anteromedial BNST, it is possible that the 30% of neurons that did not respond to both stimulation locations were primarily located outside of the anteromedial BNST. The first sentence of the discussion and figure 3B suggests the recordings were done in the anteromedial BNST, but this is not made clear throughout the manuscript. If the recordings were primarily done within the anteromedial BNST, the results need to be discussed in relation to past literature on the role of the anteromedial BNST in anxiety (Dunn, 1987; Haugler et al, 2013).

3. The data presented in this manuscript are not adequately discussed. There are three observations easily made from looking at the data presented that are not mentioned in the results or discussion:
 - a. Figure 3H suggests that AP5 not only blocks LTP of the vSUB/CA1 pathway, but also uncovers an underlying LTD at this synapse. This is not mentioned in the manuscript at all and suggests there is an NMDA-dependent LTP and NMDA-independent LTD competing at the synapse.
 - b. In Figure 3I, it seems as though the magnitude of the LTD of the ILCx pathway is significantly greater with AP5 than in the control condition, suggesting may be competing LTP and LTD processes at both synapses. This needs to be mentioned in the results and discussed in the discussion section.
 - c. The x axis for Figure 1F does not allow the reader to see the difference in the latency to response to stimulation. It would be more helpful to see only 25 - 50 msec on either side of stimulation. In addition to this more zoomed in view, it would be beneficial to show activity over the course of 500 msec on either side of the stimulation. Supplementary figure 2 shows the response to stimulation over the course of 1 second. From this view, we can see that there is a ~200 msec period of inhibition of spontaneous firing after stimulation. This is never mentioned in the manuscript. Do all neurons experience this delayed inhibition of firing? This delayed inhibition of spontaneous firing could be due to GABAergic network activity. It would be beneficial to show the wider time period in Figure 1 as well.
4. There is a general lack of clarity in the methods throughout the manuscript.
 - a. The description of the HFS used is not adequate. A figure to describe the HFS should be included.
 - b. The description of the calculation of the Rmag is not clear. For example, the meaning of "baseline" is unclear.
 - c. The time line outlined in supplementary figure 3 should be included in figure 4 of the manuscript. As written now, it is not clear that the behavioral test is performed over a week after the HFS recording.

Minor Concerns:

1. The title does not accurately convey the data presented in the article. The manuscript does not look at the BNST in its entirety and there is little evidence presented to support the idea that the LTP and LTD occurring at the different synapses is a result of homeostatic plasticity.
2. The significant difference in the latency to the onset of the stimulation response between the two inputs is reported but not discussed. The authors need to explore the significance of a 3 msec difference in latency.
3. The insets in Figure 1 cover up some of the information in the raster plots.
4. The Y axis for the raster plots is inappropriately labeled as "Raster (spikes)". The Y axis of this plot indicates the trial number and should be changed to reflect that.
5. Panel B in Figure 2 does not provide any additional information and should be removed. Instead of the current panel B, it would be beneficial to see a low magnification image of the electroporated cells in the BNST.

6. In supplementary figure 2, data is shown for the effect of HFSILCx on Rmag of the ILCx pathway. This figure should also show the effect of the HFSILCx on the vSUB/CA1 pathway.

Reviewer #1 (Remarks to the Author):

Glangetas et al set out to answer a question that is very important, but often ignored in the field's pursuit of linking synapses, circuits and behavior. Specifically, how does one reconcile changes in synaptic strength at one set of synapses with stability of firing in target neurons? Does this require some type of adaptive response at other synapses? Here they address this question beautifully using a series of elegant and challenging approaches. First they show that single neurons in BNST receive inputs from both ventral subiculum/CA1 neurons and neurons in the infralimbic cortex. Then, they show that if one set of synapses (vsub/CA1-BNST) is potentiated in vivo, then the other synapses (iLCtx-BNST) are depressed. This, they suggest indicates a type of homeostatic plasticity. This part of the manuscript is well done, but I do have some questions that, if addressed appropriately, would help strengthen this part of the paper. The authors then ask whether this plasticity has a behavioral consequence. In this part of the manuscript they fall short. In fact, there is little support for the authors assertions that, "Interplay between an NMDA-receptor-dependent homosynaptic long-term potentiation and an NMDA-receptor-independent heterosynaptic long-term depression is instrumental for anxiolysis."

Major Concerns

1) Although the authors show that potentiation at the vsub/CA1-BNST connection does promote anxiolysis (albeit it is quite modest) they fail to show any link between homeostatic changes in BNST and this behavior. Can the authors provide any evidence to support the idea that both potentiation at the vsub/CA1 pathway AND depression in the iLCx pathway are necessary for anxiolysis?

We thank the reviewer for his/her valuable and constructive comments challenging the hypothesis that both potentiation at the vsub/CA1 pathway and depression in the iLCx pathway were necessary for anxiolysis. We now provide a clear interpretation of our key experiment demonstrating that intra-BNST AP5 maintains and even potentiates LTD (Figure 3 f-g), but prevents the anxiolytic effect (Figure 4d). This experiment demonstrates that depression in the iLCx pathway is not necessary for anxiolysis. The title, the abstract and the manuscript were revised because we agree that there is not enough evidence to support the concept that both potentiation at the vsub/CA1 pathway and depression in the iLCx pathway were necessary for anxiolysis

2) If my understanding is correct, the HFS and subsequent assessment of plasticity were done in anaesthetized animals. The behavior, clearly required awake behaving animals.

a. Can the authors be certain that plasticity observed in anaesthetized animals can be faithfully reproduced in awake, behaving animals?

The HFS protocol we used for LTP induction has been demonstrated to be efficient in awake freely-moving animals but also triggered epileptiform activity and behavioral symptoms such as shacking in the majority of stimulated animals (Taylor C.J. et al. 2016. EJM, 43; 811-822). For ethical reasons and to avoid the confounding effect of epilepsy driven behavioral changes, HFS was delivered in anesthetized animals. This point is now clarified in the

methods.

b. Why was light-dark test conducted 8 days after HFS in vSUB/CA1? When examining plasticity in this pathway, the authors test for changes in spike probability, thirty to forty minutes after HFS. Do the authors have evidence that LTP lasts for 8 days after HFS?

The reviewer raises a very important point concerning the persistence of LTP in the BNST after HFS_{vSUB/CA1}. The reviewer 3 also raised a similar question. Persistence of LTP after HFS_{vSUB/CA1} was recently examine in the vSUB/CA1-mPFC pathway (Taylor C.J. et al. 2016. EJN, 43; 811-822). This protocol led to an LTP that was significantly above baseline at day 8 after HFS_{vSUB/CA1}. This information was added in the result section of the manuscript: “This HFS_{vSUB/CA1}, originally described by Abraham and colleagues²¹, induced an extremely long-lasting (> 8 days), robust and stable LTP in the projection from vSUB/CA1 to medial prefrontal cortex (mPFC)²²”.

In parallel, an attempt was made to assess the maintenance of the LTP across days in the projection from vSUB/CA1 to BNST. Since our *in vivo* measure of LTP is based on a change in spike probability, a pre-HFS condition is needed and prevents the possibility to evaluate, in our experimental condition, the spiking probability 8 days after the HFS. For these reason, we decided to perform an *in vivo* occlusion experiment to determine whether LTP lasts for 8 days after HFS. The hypothesis is that if the LTP decays steadily over hours and is back to baseline by day 8 post- HFS_{vSUB/CA1}, a second HFS_{vSUB/CA1} protocol should be able to trigger a novel LTP. To test this hypothesis, 6 rats were stimulated using an HFS_{vSUB/CA1} at D0 and were tested for a second HFS_{vSUB/CA1} at D8. In this occlusion condition, HFS_{vSUB/CA1} at D8 was unable to trigger a novel LTP (confirming that the vSUB/CA1-BNST response was not return to baseline), but surprisingly trigger an LTD. These data are recapitulated on the figure below.

Figure 1: *In vivo* occlusion LTP-plasticity in the BNST 8 days after HFS in vSUB/CA1. **a.** Experimental protocol. **b.** Quantification of the mean percentage change (\pm sem) in the vSUB/CA1 spike probability. **c.** Typical PSTHs (+ stimulation trials) illustrating responses of a same single-BNST neuron before (*left*) and after (*right*) HFS_{vSUB/CA1} on the vSUB/CA1-BNST pathway. Stimulus: t0 (gray lines). Bin width: 1ms.

We think that this new set of data support our conclusion that HFS_{vSUB/CA1} generated LTP lasting at least 8 days. However, we think that these data are too preliminary to provide a clear interpretation of the mechanisms of HFS_{vSUB/CA1} driven-LTD in occlusion condition, and propose to not include this figure in the current version of the manuscript.

3) The rationale for the HFS protocol is unclear. The reference provided (Abraham et al, 1993) does not specifically provide evidence that this is physiologically relevant. I understand that protocols to induce plasticity vary widely across preparations and brain regions, so it is important the authors provide stronger support for their contention that vsub/CA1 neurons fire at hundreds of Hertz.

We appreciate the comments of the reviewer and performed an additional experiment to measure the maximal instantaneous frequency of vSUB/CA1 neurons and found that in basal condition identified vsub/CA1 neurons fire at hundreds of Hertz. This set of data is now reported in Fig. S2 and we added a phrase in the discussion emphasizing this information : “Given that vSUB/CA1 neurons fire at hundreds of Hertz in basal condition (Fig. S2), we first assessed the impact of a high-frequency stimulation (HFS) of the vSUB/CA1 (HFS_{vSUB/CA1}), applied at physiologically relevant parameters”.

4) It appears that the 'LTP' reported in the vsub/CA1 pathway is NMDAR-dependent. Interestingly, it appears that HFS in the presence of AP5 unmasks an LTD. What is the explanation for this? Also, why is the LTD in the iLCx projection even more profound in the presence of AP5?

We thank the

The reviewer raises a very important point concerning the impact of NMDA-Rs blockade on both : LTP at vSUB/CA1 inputs and LTD at ILCx inputs. The reviewer 3 also raised similar questions. Accordingly we have now developed in the discussion two interpretations for these results. Concerning the appearance of an LTD after HFS_{vSUB/CA1} in presence of intra-BNST AP5, the first interpretation is that AP5 infusion in the BNST leads to a partial blockade of the NMDA-Rs and as shown by Cumming *et al.* (Cumming *et al*, 1996, *Neuron*, 16, 825-833), a partial blockade of NMDA receptors reverses the direction of synaptic plasticity elicited by HFS. The second interpretation is that an NMDA-dependent LTP and an NMDA-independent LTD are competing at the vSUB/CA1-BNST synapses, and AP5 by blocking the LTP unmasks the LTD. Concerning the potentiation of the LTD at ILCx inputs after intra-BNST AP5, one interpretation is that HFS_{vSUB/CA1} triggers concomitant activation of NMDA and metabotropic glutamate receptors in BNST neurons, leading to a more profound LTD in the presence of AP5 (Huang and Hsu, *Neuropharmacology*, 63; 1998-1307 2012). We now extend our discussion, emphasizing these different explanations : « Interestingly, in presence of AP5 in the BNST, the direction of plasticity at the vSUB/CA1-BNST inputs elicited by HFS_{vSUB/CA1} was switch from an LTP_{vSUB/CA1} to an LTD_{vSUB/CA1} (Fig. 3F). One possibility is that AP5, locally infused in the BNST, only block partially the NMDA receptors, and this partial blockade reverses the direction of plasticity

elicited $HFS_{vSUB/CA1}$ ³⁰. An other possibility is that an NMDA-dependent LTP and an NMDA-independent LTD are competing at the vSUB/CA1-BNST synapses, and AP5 by blocking the LTP unmasks the LTD. An other unexpected result was that the intra-BNST blockade of NMDA-Rs potentiates LTD_{ILCx} elicited by $HFS_{vSUB/CA1}$ (Fig. 3F). One possibility is that is that $HFS_{vSUB/CA1}$ triggers concomitant activation of NMDA and metabotropic glutamate receptors in BNST neurons, leading to a more profound LTD in the presence of AP5³¹. Further experiments are necessary to determine the molecular mechanisms by which in the absence of NMDA-Rs stimulation, $HFS_{vSUB/CA1}$ triggers $LTD_{vSUB/CA1}$ and potentiates LTD_{ILCx} .”

5) The increase in time spent in the light following HFS is quite modest - from 28% to 32%. If the authors are to make a strong case for 'LTP' in this pathway being anxiolytic, additional behavioral tests are warranted.

The point raised by the reviewer 1 was crucial to better understand the power of the anxiolytic effect of the $HFS_{vSUB/CA1}$ (i.e. point 1 Reviewer 3). In our experimental condition (normal situation), the increase in time spent in the light following $HFS_{vSUB/CA1}$, was from 27.5% to 36.8% (and not 4% as mentioned by the reviewer). For this reason, we decided to perform additional behavioral experiments examining the anxiolytic effect of the $HFS_{vSUB/CA1}$ in an anxiogenic situation. In the light-dark test, the light serves as an anxiogenic stimulus. Thus, in order to increase the anxious state of the rats, we replicated our original experiment in an experimental condition where the intensity of the lighting was increases from 560 Lux to 1230 Lux, and where all animals were restrained for 5 minutes before the light-dark test. As expected, sham animals decreased the percentage of time spent in the light compartment (27.5% in basal situation to 10.1% in anxiogenic situation). $HFS_{vSUB/CA1}$ triggers a similar amplitude of anxiolysis ($\approx 10\%$ increase in time spent in the light compartment) in normal rats and in anxious rats. This set of data is now reported in Fig. 4 and we added a paragraph in the discussion emphasizing the efficiency of the $HFS_{vSUB/CA1}$ in normal and anxiogenic situation. We are thankful to the reviewers since this experiment allows us to provide clear interpretations of our original behavioral results and draw more convincing conclusions about the impact of the $HFS_{vSUB/CA1}$ anxiogenic situation.

6) I'm curious about why there is not a reciprocal form of plasticity if HFS is delivered to the iLCx, rather than vsub/CA1. The authors mention this briefly, but do not discuss it further.

As mentioned by the reviewer in his/her comment 3, protocols to induce plasticity vary widely across preparations and brain regions. The HFS protocol we used to elicit plasticity has been originally used in the hippocampus (Abraham et al. 1993) and mimicked the firing discharge pattern of vSUB/CA1 neurons (new supplementary figure S2 a -c). We now added a phrase in the discussion: “This HFS protocol was efficient to trigger plasticity in the BNST when applied in the vSUB/CA1, but not in the ILCx (Fig. S3). This is probably due to the fact that vSUB/CA1 is one of the few major output structures of the hippocampal formation and transmits learning and memory-related signals in a high-frequency bursting mode (Fig S2) repeated at a low frequency (0.5-2 Hz)²⁷. »

Reviewer #2 (Remarks to the Author):

This is an interesting report demonstrating the anatomical convergence and physiological interaction of ventral hippocampal and prefrontal cortical inputs on single BNST neurons. The authors find that high-frequency stimulation of ventral hippocampal afferents in the BNST causes an NMDA-receptor dependent long-term potentiation, as well as an NMDA-independent homosynaptic depression at IL afferents. NMDA receptor antagonists in the BNST dampened the anxiolytic effects of ventral hippocampal stimulation. These are intriguing findings, but there are some important questions concerning the results as described:

1) which region of the BNST exhibited converging hippocampal and prefrontal afferents? FigS1 suggests convergence in areas both dorsal and ventral to the anterior commissure--where were the recordings obtained? The BNST is quite diverse--did the physiological properties described differ according to the region from which the recordings were made?

We thank the reviewer for his valuable and constructive comments. We have now performed a cartography to plot the location of the BNST neurons tested for plasticity, according to their response to the ILCx, the vSUB or both. All the cells displaying plasticity were localized the anteromedial part of the BNST (amBNST). The term « amBNST » is now used throughout the manuscript. This set of analysis is now reported in Fig. 3B.

2) Ventral hippocampal stimulation might have induced plasticity in other efferents including the PFC and BLA. In fact, individual neurons in the VH project to both regions and have been implicated in regulating fear. Although the effects of stimulation were dampened by BNST APV, is it possible that these other circuits play a role?

We totally agree with reviewer comment and acknowledge this possibility in the discussion: *“Finally, we can not exclude that $HFS_{vSUB/CA1}$ also trigger plasticity in the mPFC²³ or the basolateral amygdala³³, but we provide behavioral evidence that vSUB/CA1-driven NMDA-R-dependent LTP in the BNST triggers anxiolytic-like effects.”*

3) Tetanic stimulation of the ventral hippocampus was found to have an anxiolytic effect in the light:dark box, which was attenuated by NMDA antagonists in the BNST. Much of the lesion literature suggests that VH activation is anxiogenic. How do the authors account for the anxiolytic effects of titanic stimulation of VH relative to the larger literature that would suggest the opposite would be true?

We agree with reviewer comment that the anxiolytic effect we observed after $HFS_{vSUB/CA1}$ appears to contradict data of the lesion literature. However, the molecular compensatory adaptation occurring in the vSUB-lesioned animals (i.e. increases in GAD65 and GAD67 expression) make the comparison difficult with the synaptic adaption triggered by $HFS_{vSUB/CA1}$. However, based on lesion literature, important future directions where timely inactivation of inputs to the BNST using opto- or chemogenetic tools may be critical for changing network output, physiological manifestations of anxiety and anxiety-associated disorders. These interesting perspectives are now added to our

conclusion.

Reviewer #3 (Remarks to the Author):

This manuscript provides evidence for a direct excitatory projection from the infralimbic cortex (ILCx) and ventral subiculum / CA1 (vSUB/CA1) to individual neurons in the BNST. Data is provided showing that high frequency stimulation (HFS) of the vSUB/CA1 results in an NMDA-dependent homosynaptic long term potentiation (LTP) and an NMDA-independent heterosynaptic long term depression (LTD) of the ILCx input. Interestingly, HFS of the vSUB/CA1 causes a reduction in anxiety-like behavior that is blocked by AP5 in the BNST, suggesting the LTP of the vSUB/CA1 pathway is necessary for anxiolysis. This study provides data that significantly furthers our knowledge of the circuitry of the BNST, however, the representation and interpretation of the data is significantly lacking in both clarity and depth.

Major Concerns:

1. A major concern with this manuscript is the lack of a direct cause-effect relationship between the LTP and LTD due to the HFS and the behavioral changes observed. The authors should be cautious not to overstate their conclusions. The data shown suggests that HFS of the vSUB/CA1 pathway caused a decrease in anxiety-like behavior more than a week later. However, the authors directly relate the LTP and LTD that occurs immediately after the HFS with the change in behavior eight days later with no data to support this conclusion.

As mentioned in the point 2b of the Reviewer 1, we have now discussed an important study demonstrating the persistence of LTP after HFS_{vSUB/CA1} in the vSUB/CA1-mPFC pathway (Taylor C.J. et al. 2016. EJM, 43; 811-822). This protocol led to an LTP that was significantly above baseline at day 8 after HFS_{vSUB/CA1}. Moreover, we performed an *in vivo* occlusion experiment to determine whether LTP lasts for 8 days after HFS (please see response to the comment 2b of reviewer 1).

Additionally, a second behavioral anxiety test would strengthen the evidence that HFS of the vSUB/CA1 can have long-term effects on anxiety-like behavior in rats.

As mentioned in the point 5 of the Reviewer 1, we have now reported a second behavioral anxiety test that indeed strengthen the evidence that HFS of the vSUB/CA1 have long-term effects on anxiety-like behavior in rats and allow us to draw more convincing conclusions about the impact of the HFS_{vSUB/CA1} anxiogenic situation. Please see Point 5 of reviewer 1 for details.

2. The location of the recordings in the BNST needs to be made clearer. There is significant evidence that the different regions of the anterior BNST, including the oval, anterolateral, anteromedial, and ventral BNST, have different effects on anxiety-like behavior in rodents (Dunn, 1987; Haufler et al, 2013; Kim et al,

2013). If recordings were from throughout the anterior BNST, it is unclear how LTP of the vSUB/CA1 synapses could cause a reduction in anxiety-like behavior when optogenetic and chemogenetic inhibition of some of these neurons is known to decrease anxiety-like behavior (Kim et al, 2013; Pleil et al, 2015). The proportion of neurons that respond to each stimulation should be described in terms of their location in the BNST. For example, if the majority of recordings, but not all, were performed in the anteromedial BNST, it is possible that the 30% of neurons that did not respond to both stimulation locations were primarily located outside of the anteromedial BNST.

We thank the reviewer for his valuable and constructive comments. We have now performed a cartography to plot the location of the BNST neurons tested for plasticity, according to their response to the ILCx, the vSUB or both. All the cells displaying plasticity were localized the anteromedial part of the BNST (amBNST). The term »amBNST » is now used throughout the manuscript. This set of analysis is now reported in Fig. 3B.

The first sentence of the discussion and figure 3B suggests the recordings were done in the anteromedial BNST, but this is not made clear throughout the manuscript. If the recordings were primarily done within the anteromedial BNST, the results need to be discussed in relation to past literature on the role of the anteromedial BNST in anxiety (Dunn, 1987; Haufler et al, 2013).

We thank the reviewer for bringing these references to our attention. We have now included this important literature in the discussion: *“ This is in line with pioneer studies showing that changing the activity in the amBNST has a direct impact on the perception of aversive contextual stimuli³² or production of stress hormones⁵.”*

3. The data presented in this manuscript are not adequately discussed. There are three observations easily made from looking at the data presented that are not mentioned in the results or discussion:

- a. Figure 3H suggests that AP5 not only blocks LTP of the vSUB/CA1 pathway, but also uncovers an underlying LTD at this synapse. This is not mentioned in the manuscript at all and suggests there is an NMDA-dependent LTP and NMDA-independent LTD competing at the synapse.**
- b. In Figure 3I, it seems as though the magnitude of the LTD of the ILCx pathway is significantly greater with AP5 than in the control condition, suggesting may be competing LTP and LTD processes at both synapses. This needs to be mentioned in the results and discussed in the discussion section.**

As mentioned in the point 4 of the Reviewer 1, we have now discussed this very important point concerning the impact of NMDA-Rs blockade on both : LTP at vSUB/CA1 inputs and LTD at ILCx inputs. Please see response to Reviewer 1 (point 4.) for details.

c. The x axis for Figure 1F does not allow the reader to see the difference in the latency to response to stimulation. It would be more helpful to see only 25 - 50 msec on either side of stimulation. In addition to this more zoomed in view, it would be beneficial to show activity over the course of 500 msec on either side of the stimulation.

We are thankful to the reviewer for his/her suggestion. We have changed the scales of the PSTHs accordingly.

Supplementary figure 2 shows the response to stimulation over the course of 1 second. From this view, we can see that there is a ~200 msec period of inhibition of spontaneous firing after stimulation. This is never mentioned in the manuscript. Do all neurons experience this delayed inhibition of firing? This delayed inhibition of spontaneous firing could be due to GABAergic network activity. It would be beneficial to show the wider time period in Figure 1 as well.

We thank the reviewer for this comment. We verified all the neurons responding to both inputs and tested for plasticity and only 3 out of 25 displayed this robust inhibition. The other 22 neurons have a slow basal activity, and no evoked inhibition was detectable. We changed the PSTH accordingly to be representative of the slow activity of the BNST neurons.

4. There is a general lack of clarity in the methods throughout the manuscript.

We thank the reviewer for this comment and revisions have been made accordingly (in red in the manuscript).

a. The description of the HFS used is not adequate. A figure to describe the HFS should be included.

We have now clarified in the methods and in a supplemental figure (Fig S2) the description of the high frequency stimulation protocol. It consisted in 50 trains (500 pulses at 400 Hz, 250 μ s duration pulse) presented as bursts of 5 trains. The frequency of the five trains was 1 Hz. Each burst of 5 trains was presented 5 times at 1 min interval.

b. The description of the calculation of the R_{mag} is not clear. For example, the meaning of "baseline" is unclear.

We thank the reviewer for this comment and revisions have been made accordingly: *"Baseline activity was calculated on each PSTH, during the 500 ms preceding the stimulation. For each PSTH, R_{mag} values for excitation were calculated according to: Excitation $R_{mag} = (\text{number of spikes in excitatory epoch}) - (\text{mean number of spikes per baseline bins} \times \text{number of bins in excitatory epoch})$."*

c. The time line outlined in supplementary figure 3 should be included in figure 4 of the manuscript. As written now, it is not clear that the behavioral test is performed over a week after the HFS recording.

We thank the reviewer for this comment and revisions have been made accordingly.

Minor Concerns:

1. The title does not accurately convey the data presented in the article. The manuscript does not look at the BNST in its entirety and there is little evidence presented to support the idea that the LTP and LTD occurring at the different synapses is a result of homeostatic plasticity.

We thank the reviewer for this comment and revisions have been made accordingly. The new title is : “*NMDA receptors-dependent plasticity in the Bed Nucleus of the Stria terminalis triggers long-term anxiolysis*”.

2. The significant difference in the latency to the onset of the stimulation response between the two inputs is reported but not discussed. The authors need to explore the significance of a 3 msec difference in latency.

We thank the reviewer for this comment and this point has been discussed accordingly: “*Considering the differences in the straight-line distances between the ILCx-BNST (≈ 4 mm) and vSUB/CA1-BNST (≈ 6 mm), the 3 ms difference in the latency to the onset of the stimulation response between the two inputs is probably not supported by a difference in conduction velocity, but by the length of the axonal projections.* “

3. The insets in Figure 1 cover up some of the information in the raster plots.

We thank the reviewer for this comment and revisions have been made accordingly.

4. The Y axis for the raster plots is inappropriately labeled as "Raster (spikes)". The Y axis of this plot indicates the trial number and should be changed to reflect that.

We thank the reviewer for this comment and revisions have been made accordingly.

5. Panel B in Figure 2 does not provide any additional information and should be removed.

We thank the reviewer for this comment and revisions have been made accordingly.

Instead of the current panel B, it would be beneficial to see a low magnification image of the electroporated cells in the BNST.

We thank the reviewer for this comment and revisions (removal of the PSTH) have been made accordingly. Unfortunately we were unable to detect the starter cell in the BNST, most probably because rabies virus infection will eventually induce cytotoxicity in infected neurons. This limitation is important and has been previously discussed in a recent publication of Dr. E. Callaway. We now included this point of discussion and the associated reference in the manuscript : « *Five days after electroporation, retrogradely infected neurons were detected in the ILCx and in the vSUB/CA1 (Fig.2c). Unfortunately, starter cells were not detected in the ambNST, most probably because rabies virus infection will eventually induce cytotoxicity in infected neurons* ²¹. “

6. In supplementary figure 2, data is shown for the effect of HFSILCx on Rmag of the ILCx pathway. This figure should also show the effect of the HFSILCx on the vSUB/CA1 pathway.

We appreciate the comments of the reviewer and performed an additional experiment to measure the effect of HFS_{ILCx} on vSUB/CA1-BNST pathway (Fig. S3). No plasticity was evoked on vSUB/CA1-BNST after HFS_{ILCx} .

Reviewers' comments:

Reviewer #1 (Remarks to the Author):

I appreciate the authors' very careful responses and additional data. They do a commendable job of addressing many of the key issues. The additional experiments showing firing of vSub/CA1 neurons offers support for delivering high frequency trains, but given that the example they show rarely shows instantaneous frequencies of > 125 Hz, I would prefer that the authors remove the phrase, "applied at physiologically relevant parameters".

The explanation that AP5 at lower concentrations can reverse the polarity of the plasticity seems sensible. The Cumming et al paper cited is appropriate. In addition, work by Bains et al, Nat Nsci, 1999 demonstrates that network output can be reversibly altered by inducing synaptic depression with low doses of AP5. This offers strong support for the authors' model. Less convincing is the argument that, "NMDA-dependent LTP and an NMDA-independent LTD are competing at the vSUB/CA1-BNST synapses, and AP5 by blocking the LTP unmasks the LTD." This seems like conjecture and is not needed in the discussion.

The additional behavioral test provided by the authors is another light-dark test using different experimental parameters. This, in my estimation, is not a different test of anxiety. What about an open field? Elevated plus maze? Elevated zero maze? One of these additional tests would have strengthened the paper considerably.

It is a pity that, in the end, the homeostatic synaptic changes were not necessary for the behavioral changes. Nevertheless, the observations remain strong and the work is thoroughly and carefully done.

The reference list needs attention. There was clearly an issue when it was compiled and appended to the manuscript.

Reviewer #2 (Remarks to the Author):

The authors have satisfactorily addressed my concerns. It is a timely and exciting report.

Reviewer #3 (Remarks to the Author):

In the revised manuscript Glangetas, Georges, and colleagues have gone a long way to address many of the concerns raised in the original review and, consequently, the revised manuscript is considerably improved.

Having said that, the question of cause-effect relationship between LTP induction in the vSUB-ambNST pathway and the anxiolytic behavioral response has not been definitively addressed, despite the experiments showing occlusion of LTP induction in the ambNST following HFS stimulation of the vSUB pathway on D8. Indeed, the appearance of an LTD response to HFS in the same pathway on D8 suggests that multiple overlapping pathways may be recruited to modulate ambNST activity and any subsequent behavior.

Moreover, there is still a concern about the physiological relevance of the HFS paradigm. The authors contend that ventral subiculum neurons fire at "hundreds of Hz" at baseline. While it is true that the instantaneous firing rate of these neurons can reach > 100 Hz, this firing rate is not maintained for any significant duration (milliseconds) and, hence, cannot be directly compared to the HFS paradigm. Although the authors have added an additional behavioral manipulation to test the anxiolytic response

to vSUB stimulation, the test was just another variation of the light-dark test. Inclusion of an open field test and/or elevated plus maze would have been preferable to determine the effects on trait anxiety versus stimulus-induced anxiety. Lastly, the revised manuscript has a lot of grammatical errors that should be corrected.

Reviewer #1 (Remarks to the Author):

I appreciate the authors' very careful responses and additional data. They do a commendable job of addressing many of the key issues. The additional experiments showing firing of vSub/CA1 neurons offers support for delivering high frequency trains, but given that the example they show rarely shows instantaneous frequencies of > 125 Hz, I would prefer that the authors remove the phrase, "applied at physiologically relevant parameters".

We thank the reviewer for his/her valuable and constructive comments and have now removed from the manuscript all allusions of our electrical stimulation being applied at physiologically relevant parameters (Result: p7 line 4; Discussion p9 line 7 and page 11 line 11).

The explanation that AP5 at lower concentrations can reverse the polarity of the plasticity seems sensible. The Cumming et al paper cited is appropriate. In addition, work by Bains et al, Nat Nsci, 1999 demonstrates that network output can be reversibly altered by inducing synaptic depression with low doses of AP5. This offers strong support for the authors' model.

We thank the reviewer for bringing the work of Bains et al (1999) to our attention. This references in now added to our discussion.

Less convincing is the argument that, "NMDA-dependent LTP and an NMDA-independent LTD are competing at the vSUB/CA1-BNST synapses, and AP5 by blocking the LTP unmasks the LTD." This seems like conjecture and is not needed in the discussion.

We agree and removed this conjuncture from the discussion.

The additional behavioral test provided by the authors is another light-dark test using different experimental parameters. This, in my estimation, is not a different test of anxiety. What about an open field? Elevated plus maze? Elevated zero maze? One of these additional tests would have strengthened the paper considerably.

The point raised by the reviewer 1 was crucial to better understand the power of the anxiolytic effect of the $HFS_{vSUB/CA1}$ (i.e. Reviewer 3). For this reason, we decided to perform additional behavioral experiments examining the anxiolytic effect of the $HFS_{vSUB/CA1}$ using the classical Elevated plus maze. The anxiolytic effect observed in the light-dark test was similar to the one observed in the elevated plus maze. This set of data is now reported in Fig. 4 and in the result. We are thankful to the reviewers since this experiment allows us to provide clear interpretations of our original behavioral results and draw more convincing conclusions about the impact of the $HFS_{vSUB/CA1}$.

The reference list needs attention. There was clearly an issue when it was compiled and appended to the manuscript.

Thanks to the reviewer, this has been corrected.

Reviewer #2 (Remarks to the Author):

The authors have satisfactorily addressed my concerns. It is a timely and exciting report.

We thank the reviewer for his/her valuable and constructive comments during these two rounds of revision.

Reviewer #3 (Remarks to the Author):

In the revised manuscript Glangetas, Georges, and colleagues have gone a long way to address many of the concerns raised in the original review and, consequently, the revised manuscript is considerably improved.

Having said that, the question of cause-effect relationship between LTP induction in the vSUB-ambNST pathway and the anxiolytic behavioral response has not been definitively addressed, despite the experiments showing occlusion of LTP induction in the ambNST following HFS stimulation of the vSUB pathway on D8. Indeed, the appearance of an LTD response to HFS in the same pathway on D8 suggests that multiple overlapping pathways may be recruited to modulate ambNST activity and any subsequent behavior.

Moreover, there is still a concern about the physiological relevance of the HFS paradigm. The authors contend that ventral subiculum neurons fire at “hundreds of Hz” at baseline. While it is true that the instantaneous firing rate of these neurons can reach > 100 Hz, this firing rate is not maintained for any significant duration (milliseconds) and, hence, cannot be directly compared to the HFS paradigm.

We thank the reviewer for his/her valuable and constructive comments. We agree that the switch of polarity of plasticity observed after AP5 or a second HFS protocol performed on day 8, revealed the complexity of the plastic adaptation *in vivo* and suggest multiple overlapping pathways necessary to modulate the activity of the ambNST. As mentioned in the point 1 of the Reviewer 1, we have now removed from the manuscript all allusions of our electrical stimulation being applied at physiologically relevant parameters (Result: p7 line 4; Discussion p9 line 7 and page 11 line 11).

Although the authors have added an additional behavioral manipulation to test the anxiolytic response to vSUB stimulation, the test was just another variation of the light-dark test. Inclusion of an open field test and/or elevated plus maze would have been preferable to determine the effects on trait anxiety versus stimulus-induced anxiety.

The reviewer raises a very important point concerning the necessity of duplicating the key data of $HFS_{vSUB/CA1}$ triggering anxiolysis, with a second behavioral assay. The reviewer 1 also raised a similar question. For this reason, we decided to perform additional behavioral experiments examining the anxiolytic effect of the $HFS_{vSUB/CA1}$ using the classical Elevated plus maze. The anxiolytic effect observed in the light-dark test was similar to the one observed in the elevated plus maze. This set of data is now reported in Fig. 4 and in the result. We are thankful to the reviewers since this experiment allows us to provide clear interpretations of our original behavioral results and draw more convincing conclusions about the impact of the $HFS_{vSUB/CA1}$.

Lastly, the revised manuscript has a lot of grammatical errors that should be corrected.

The manuscript has now been carefully edited for grammatical errors.

REVIEWERS' COMMENTS:

Reviewer #1 (Remarks to the Author):

The authors have addressed my comments.

Reviewer #3 (Remarks to the Author):

The authors have adequately addressed all of the concerns raised in the previous review and the manuscript is significantly improved as a consequence.

Several grammatical should be corrected :-

page 4, line 60 delete s in informations

page 4, line 61 change are to is

page 4, line 68 insert "the" BNST integrates.....

page 9, line 175 delete s in informations

REVIEWERS' COMMENTS:

Reviewer #1 (Remarks to the Author):

The authors have addressed my comments.

We thank the reviewer for his/her valuable and constructive comments during these three rounds of revision.

Reviewer #3 (Remarks to the Author):

The authors have adequately addressed all of the concerns raised in the previous review and the manuscript is significantly improved as a consequence.

Several grammatical should be corrected :-

page 4, line 60 delete s in informations

page 4, line 61 change are to is

page 4, line 68 insert "the" BNST integrates.....

page 9, line 175 delete s in informations

We thank the reviewer for his/her valuable and constructive comments that improved the quality of the paper. The manuscript has now been carefully edited for grammatical errors.